# Early Growth Responses of *Larix kaempferi* (Lamb.) Carr. Seedling to Short-Term Extreme Climate Events in Summer

Nam-Jin Noh [1], Gwang-Jung Kim [2], Yowhan Son [2] and Min-Seok Cho [3,*]

1    Department of Forest Resources, Kangwon National University, Chuncheon 24341, Korea; njnoh@kangwon.ac.kr

2    Department of Environmental Science and Ecological Engineering, Korea University, Seoul 02841, Korea; kimgj20@korea.ac.kr (G.-J.K.); yson@korea.ac.kr (Y.S.)

3    Forest Technology and Management Research Center, National Institute of Forest Science, Pochoen 11186, Korea

\*    Correspondence: mscho1143@korea.kr; Tel.: +82-031-540-1143

**Abstract:** Extreme climate events such as heat waves, drought, and heavy rainfall are occurring more frequently and are more intense due to ongoing climate change. This study evaluated the early growth performance of one-year-old *Larix kaempferi* (Lamb.) Carr. seedlings under open-field extreme climate conditions including experimental warming and different precipitation regimes. We recorded the survival rate, root collar diameter, height, biomass, shoot-to-root ratio, and seedling quality index using nine treatments (three temperature levels, i.e., control, warming by 3 °C and by 6 °C, × three precipitation levels, i.e., control, drought, and heavy rainfall) in July and August 2020. The survival rate of seedlings did not differ between treatments, showing high values exceeding 94% across treatments. The measured shoot height was largest under warming by 3 °C and high rainfall, indicating that moderate warming increased seedling height growth in a moist environment. Heavy rainfall decreased stem volume by 21% and 25% under control and warming by 6 °C treatments, respectively. However, drought manipulation using rain-out shelters did not decrease the growth performance. Overall, extreme climate events did not affect the survival rate, biomass, shoot-to-root ratio, and seedling quality index of *L. kaempferi*. We thus conclude that, regarding growth responses, *L. kaempferi* seedlings may be resistant to short-term extreme warming and drought events during summer.

**Keywords:** biomass accumulation; climate change; coniferous species; open-field experiments; precipitation; larch; survival rate; warming

## 1. Introduction

Climate change results in an increased frequency and higher intensity of extreme weather events such as heat waves, drought, and heavy rainfall [1–3]. Persistent global warming has also caused extremely high temperatures, excessive precipitation, and severe drought events [3–5]. Climate models report that the global mean temperature rise could exceed 4 °C in the 21st century and the extreme climates associated with different global warming scenarios have been extensively investigated [3]. Extreme climatic events have been defined in different ways, based on statistical quantification of climatic variables or synthetic extremes of both driving and response variables [6,7]. The Intergovernmental Panel on Climate Change defined an extreme climatic event as an event being rarer than the 10th or 90th percentile of climate events within its statistical frequency distribution at a particular place over a certain period of time [1,7]. Such unprecedented climate events and uncertain future climate projections may affect tree growth, mortality, and forest structure and functioning [6–10]. Given the effects of extreme climate events on plants, these processes may reduce forest productivity, so it is crucial to predict the responses of plants to extreme climate events.

Many plants are resilient to heat stress and moisture deficit to some extent; however, extreme climate events irreversibly damage various physiological traits and mechanisms of plants [11,12]. Extreme heat deteriorates the photosynthetic processes, chlorophyll functioning, and biomass production of woody plants [11,13], and extreme drought markedly affects physiological traits and reduces tree biomass, which, however, depends on external conditions throughout the year [14–17]. Moreover, excessive rainfall may also decrease plant productivity under extreme rainfall due to reduced photosynthetic and stomatal conductance under waterlogged conditions [18]. However, it is unclear whether tree mortality occurs only due to an excessively wet environment [19]. The vulnerability of plants may increase when such extreme events occur simultaneously. For instance, extreme heat in combination with drought increases tree mortality [20]. Therefore, multifactor experiments are crucial for determining the interaction effects of simultaneous extreme climate events [15].

The early growth performance of small or young seedlings planted in open fields can be particularly affected by extreme temperatures and precipitation [21,22]. Various adverse effects of unpredictable climate factors on seedling quality, which can be assessed by measuring numerous morphological and physiological traits, have been observed in many nursery and silvicultural operations [23]. Furthermore, seedling growth depends on seedling quality [24], and whether seedlings can survive and thrive under extreme climatic conditions in nurseries is a key factor in determining the supply of seedlings and, thus, the success of plantations and restorations [17,25,26]. Despite substantial information on the production of high-quality seedlings, the growth responses of seedlings to extreme climate events in nurseries and at planting sites under changing climates are not yet comprehensively understood [17,27–29].

In this study, we used the Japanese larch, *Larix kaempferi* (Lamb.) Carr., which is one of the most popular plantation tree species due to its high economic value, wood quality, and restoration potential in Asia [17,30]. The plantation areas of *L. kaempferi* gradually increased approximately 10-fold over the past decade across South Korea, and the planted area in 2019 accounted for approximately 20% (4559 ha) of the total plantation area [31]. *Larix kaempferi* is an important fast-growing deciduous coniferous species encircling the Northern Hemisphere, due to its disease and cold resistance compared to other species [17,30–34]. In general, drought stress is considered a main factor causing a decline in larch tree species [10,32,33], whereas saturated soil water conditions may also adversely affect photosynthetic activity and stomatal conductance [18]. A few previous studies suggested a high risk of growth cessation in similar larch species such as *Larix principis-rupprechtii* Mayr planted across large areas in north-central China during extreme drought [10], and quantitative changes in the distribution and productivity of *L. kaempferi* under future climate change were predicted [34]. However, the growth responses of *L. kaempferi* seedlings to multiple factors of extreme climate have not been examined, even though this would be required to understand how climate change will affect successful planting in the future.

Our previous studies showed that physiological traits such as photosynthetic activity, stomatal conductance, and the transpiration rate of two-year-old *L. kaempferi* were reduced due to warming by 3 °C in the second year of the warming treatment [35], and warming by 3 °C combined with reduced precipitation increased the mortality of one-year-old *L. kaempferi* seedlings due to drought-induced heat stress [26]. However, the impact of multiple factors of extreme climate events on the early growth performance, biomass allocation, and seedling quality of *L. kaempferi* has not yet been documented. Therefore, the objective of this study was to determine the survival rate and growth performance of larch seedlings under experimental warming and precipitation conditions. We hypothesized that moderate warming would increase early growth performance regarding height, root collar diameter (RCD), biomass accumulation, and survival rate, whereas extreme drought, excessive precipitation, and extremely high temperatures should decrease the growth performances and hence seedling quality index and survival rate.

## 2. Materials and Methods

### 2.1. Experimental Design

This study was conducted at an experimental tree nursery located in the Forest Technology and Management Research Center, Pocheon, Korea (37°45′39″ N, 127°10′13″ E, 106 m a.s.l.). The annual mean air temperature and precipitation in this area are typically 10.2 °C and 1365 mm, respectively (1997–2019) [36]. In April 2020, 27 experimental plots of 1.5 × 1.0 m were established (Figure 1a). In total, 88 one-year-old *L. kaempferi* seedlings were planted with 11-cm intervals between seedlings in each plot containing a homogeneous sandy loam soil (70% sand, 20% silt, and 10% clay), following the guidelines for nursery practices (Figures 1b and S1b) [37]. The seeds obtained from a seed orchard located in Anmyeondo (36°29′ N, 126°23′ E, 40–50 m a.s.l.) of the National Forest Seed Variety Center were grown in the container tree nursery at the Forest Technology and Management Research Center. The annual mean air temperature and precipitation in the seed orchard are 13 °C and 1380 mm, respectively [36]. There was no difference in the initial RCD and height of the planted seedlings between pretreatment plots ($p > 0.05$, Table S1). The plots were arrayed with buffer zones of 0.5 m to the external boundary and 1.5 m between plots to prevent treatments from affecting each other (Figure 1).

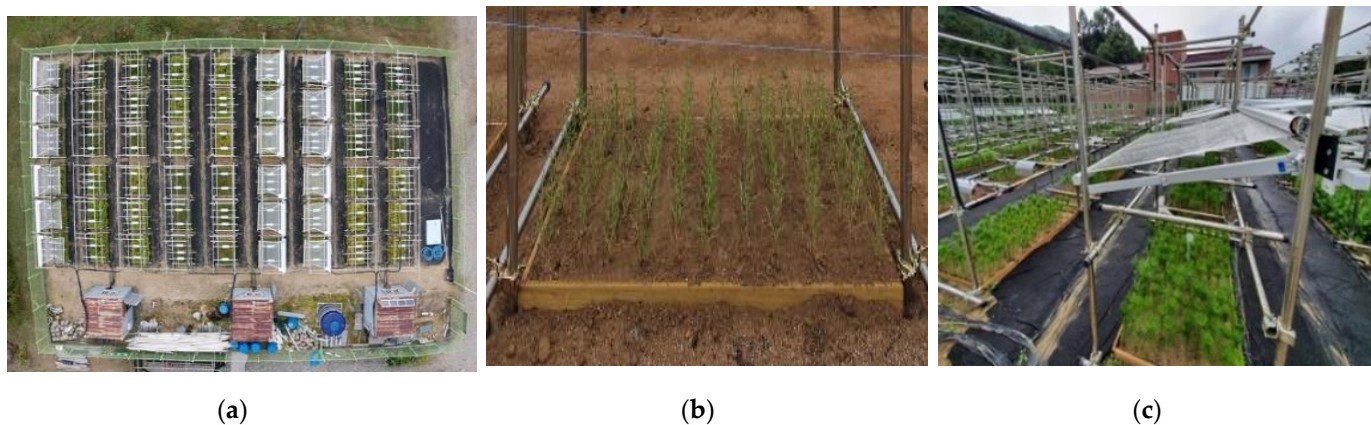

| **(a)** | **(b)** | **(c)** |
|---|---|---|

**Figure 1.** (**a**) Overview of the open-field experiment of extreme climate event manipulation; (**b**) one-year-old *Larix kaempferi* (Lamb.) Carr. seedlings (*n* = 88) planted in a plot of 1.5 × 1.0 m; and (**c**) climate manipulation systems using infrared heating lamps for warming, rain shelters for drought, and irrigation spraying for heavy rainfall treatments.

In July and August 2020, the experimental plots were subjected to nine treatments (three temperature levels [TC: ambient, T3: warming by 3 °C, T6: warming by 6 °C] × three precipitation levels [DR: drought, PC: ambient, HR: heavy rainfall]), using three replicates per treatment (Figure 2). The target temperatures for T3 and T6 were manipulated based on the 90th and 99th percentiles of the daily maximum temperature at the study site during the reference period (1961–2019) [38]. For the DR treatment, we considered the longest consecutive days with rainfall <1 mm during the reference period [39], and the HR treatment was produced using the 95th percentile of daily precipitation [40,41]. Consequently, the DR treatment lasted for nine days during the reference period, and the threshold of HR treatment was 113 mm per day [42].

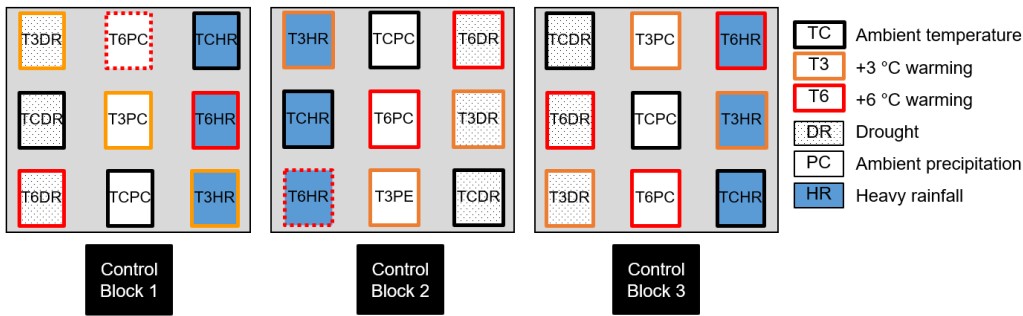

**Figure 2.** Open-field experimental design of extreme climate events regarding temperature and precipitation using *L. kaempferi* seedlings.

Based on the analysis of extreme climate scenarios, we elevated the soil surface temperature by 3 °C and 6 °C, respectively, for seven days, compared to the ambient temperature to produce T3 and T6 treatments. We used infrared heaters (FT-1000, Mor Electronic Heating Assoc., Comstock Park, MI, USA) mounted approximately 30 cm above the canopy of the seedlings and thermometers (SI-111, Apogee Instruments, Logan, UT, USA) connected to data loggers (CR1000X, Campbell Scientific, Inc., Logan, UT, USA) and relays (SDM-CD-16AC, Campbell Scientific, Inc.). We then excluded ambient rainfall for nine days to simulate drought, and HR plots were irrigated with 113 mm water per day every three days during the DR treatment period. To manipulate the DR plots, we used automatic rainout shelters coupled with rain detectors (Figure 1c). To simulate heavy rain, we simultaneously applied 170 L of stored water for 1.4 h per day at a pressure of 1.0 bar to each plot using rainfall simulators consisting of two spraying nozzles (Unijet D5-35, Spraying Systems Co., Wheaton, IL, USA) per plot. These manipulations were repeated twice during the experimental period (day of year 195–233).

We measured the soil temperature (°C) and volumetric soil water content (vol %) at 5 cm soil depth at the center of each plot every 30 min using soil sensor probes (CS655, Campbell Scientific, Inc., Logan, UT, USA) connected to the data loggers. An automatic weather station recorded the precipitation every hour at the study site.

### 2.2. Seedling Measurements

The survival of all seedlings was monitored during the experimental period. Measurements of RCD (in mm, at ground level) and shoot length (height, H, in mm) were made using digital calipers and folding rulers, respectively. For growth measurements, 30 of 88 seedlings per plot were selected from near the center (*n* = 90 per treatment) as the 30 seedlings were relatively well warmed by the infrared heating lamps compared to the outer seedlings (Figure S1). The measurements were made before (mid-May) and after (early October) the treatments. In October, three of the surviving seedlings per plot, grown from near the center of the plot, were harvested to measure the biomass accumulation and allocation (*n* = 9 per treatment). The harvested seedlings were divided into shoots (leaf and stem) and roots, and the samples were weighed after oven-drying at 65 °C for 96 h.

Height and RCD measurements were used to calculate the height-to-RCD ratio (H/D) and stem volume (V, cm³) of seedlings, which were approximated using the equation for an elliptical cone [43]:

$$V = \pi \times \text{RCD}^2 \times \text{H}/6. \tag{1}$$

Root (R) and shoot (S) biomass were used to calculate the total biomass (T) and shoot-to-root ratio (S/R) and seedling quality index (SQI) using the following equation [23,44,45]:

$$\text{SQI} = \text{T}/(\text{H/D ratio} + \text{T/R ratio}). \tag{2}$$

*2.3. Data Analyses*

A two-way analysis of variance was used to examine the effects of temperature and precipitation and their interactions on soil temperature and moisture, and on the survival rate, RCD, height, volume, biomass, H/D and R/S ratios, and SQI of seedlings. We excluded one replicate from the T6PC and T6HR treatments as two plots among the 27 randomized complete block-designed plots were partly malfunctional. Consequently, we analyzed the data from each treatment as replicates ($n = 60$ or 90 per treatment for RCD and height; $n = 9$ per treatment for biomass). When significant results were observed, Tukey's HSD test was used to determine significant differences in mean values between treatments. To determine the effect of precipitation treatment on tree height growth with the exclusion of the influence of RCD, a linear mixed-effects model was fitted using the 'lme' function of the 'nlme' R package [46]. The precipitation treatment was considered a fixed factor, the seedlings within blocks were a random factor, and the RCD was a covariate. Pearson correlations between height and RCD were analyzed using the 'stat_corr' function of the 'ggpubr' R package. Statistical significance is reported at $\alpha = 0.05$. All statistical analyses were performed with the available variables measured in May and October using R version 4.1.1 [47].

## 3. Results

*3.1. Effects of Temperature and Precipitation Treatments on Seedling Development*

Temperature (T) treatment significantly affected the soil temperature ($p < 0.001$), but did not affect the soil water content (Table 1). Precipitation (P) treatment also significantly affected the soil water content ($p < 0.001$), but did not affect soil temperature (Table 1). There was no interaction effect of temperature and precipitation treatments on soil temperature and water content. Mean temperatures were $22.7 \pm 0.3\,°C$, $25.5 \pm 0.2\,°C$, and $28.6 \pm 1.1\,°C$ in TC, T3, and T6, respectively, and mean soil moisture content was $7.0\% \pm 1.0\%$, $10.5\% \pm 1.5\%$, and $13.3\% \pm 3.1\%$ in DR, PC, and HR, respectively (Figure 3).

**Table 1.** Statistical significance (*p*-values) of the two-way ANOVA of the effects of experimental extreme climate events on the measured variables.

| Treatment | d.f. [1] | Growth Responses | | | | | | | |
|---|---|---|---|---|---|---|---|---|---|
| | | Survival | RCD | Height | Volume | H/D | Total Biomass | S/R [2] | SQI [3] |
| Temperature (T) | 2 | 0.305 | 0.258 | 0.270 | 0.176 | 0.373 | 0.881 | 0.419 | 0.498 |
| Precipitation (P) | 2 | 0.576 | 0.598 | <0.001 | 0.155 | <0.001 | 0.955 | 0.390 | 0.145 |
| T × P | 4 | 0.967 | 0.035 | <0.001 | <0.001 | <0.001 | 0.785 | 0.788 | 0.562 |

[1] d.f.: degree of freedom, [2] S/R: shoot to root biomass ratio, [3] SQI: seedling quality index.

The survival rate of seedlings did not differ between treatments (Table 1) and was generally high, with values exceeding 94% across treatments (Figure 4a, Table S1). There was no significant difference in the RCD and height measured in mid-May between pre-treatments (Table S1). There was no significant effect of the treatments on RCD, which showed a narrow range of 6.4 mm (T6HR) to 7.1 mm (T3HR) at the end of the experiment (Figure 4b).

The warming treatments did not affect seedling height, but an effect of precipitation and interaction effects on height were observed ($p < 0.001$, Table 1). The height of the seedlings was largest in T3HR (57.5 cm) and T6DR (56.0 cm) and smallest in TCHR (45.9 cm) and T6HR (48.6 cm; $p < 0.05$, Figure 4c).

Temperature and precipitation treatments did not affect the estimated stem volume of seedlings, but significant interaction effects on the volume were observed ($p < 0.001$, Table 1). The stem volume of seedlings was highest in T3HR (15.4 cm$^3$ seedling$^{-1}$) and lowest in TCHR and T6HR (11.0 and 10.4 cm$^3$ seedling$^{-1}$, respectively; $p < 0.05$). The seedling biomass (root, stem, leaf, and total biomass) did not differ between treatments. The total biomass of seedlings ranged from 13.24 to 15.88 g seedling$^{-1}$ (Table 2). The small

sample size may result in no significant difference in biomass accumulation between the treatments; however, stem biomass was directly correlated with stem volume across the treatments ($p < 0.001$, Figure S2).

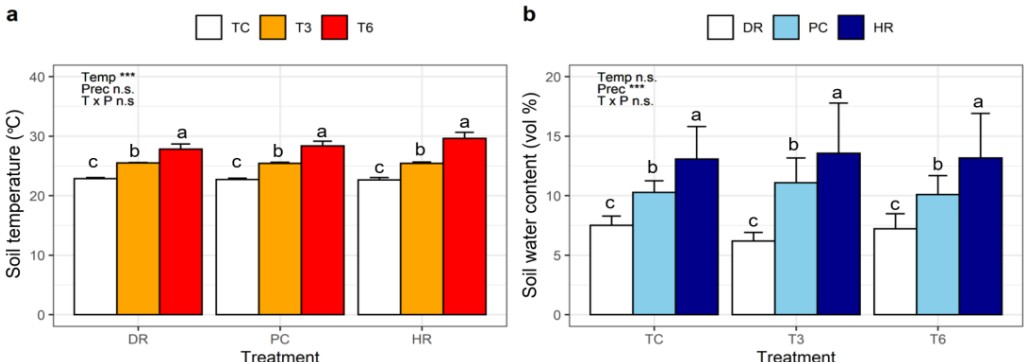

**Figure 3.** (**a**) Mean soil temperature (°C) and (**b**) mean soil water content (vol %) during the experimental period of extreme climate events. TC: ambient temperature, T3: +3 °C warming, T6: +6 °C extreme warming, DR: drought, PC: ambient precipitation, HR: heavy rainfall. Results of a two-way ANOVA are shown in each panel. Asterisks indicate significant differences between treatments; n.s.: not significant. Different letters indicate significant differences between treatment within each group factor. Error bars indicate the standard deviation of the mean ($n = 3$).

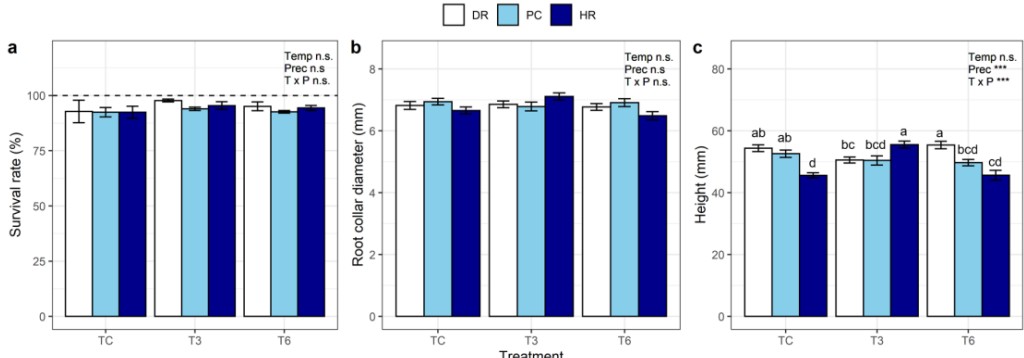

**Figure 4.** (**a**) Survival rate (%), (**b**) root collar diameter (mm), and (**c**) height (mm) of *L. kaempferi* seedlings subjected to extreme climate events, measured in October. TC: ambient temperature, T3: +3 °C warming, T6: +6 °C extreme warming, DR: drought, PC: ambient precipitation, HR: heavy rainfall. Results from two-way ANOVA are shown in each panel. Asterisks indicate significant differences between treatments; n.s.: not significant. Different letters in (**c**) indicate significant differences between treatments. Bars indicate standard errors of the mean.

### 3.2. S/R Ratios and SQI

Height was strongly correlated with RCD, but the treatments did not affect the relationship between RCD and height (Figure 5a). The shoot-to-root ratio was not significantly affected by the treatments, with a range from 2.75 to 3.42 (Figure 5b). The SQI was also not affected by the treatments (Table 2).

**Table 2.** Biomass and seedling quality index (SQI) of *L. kaempferi* (Lamb.) Carr. seedlings under extreme climate events regarding temperature and precipitation. TC: ambient temperature, T3: warming by 3 °C, T6: warming by 6 °C, DR: drought, PC: ambient precipitation, HR: heavy rainfall. Values are means ± S.D. (*n* = 9). There was no significant difference in the variables between treatments at 5% levels using a Tukey HSD test.

| Treatment | | Leaf Biomass (g) | Stem Biomass (g) | Root Biomass (g) | Total Biomass (g) | SQI |
|---|---|---|---|---|---|---|
| TC | DR | 4.06 ± 0.42 | 7.03 ± 1.27 | 3.76 ± 0.74 | 14.87 ± 2.00 | 1.23 ± 0.23 |
|    | PC | 4.80 ± 0.90 | 7.17 ± 1.43 | 4.21 ± 1.07 | 15.88 ± 3.25 | 1.37 ± 0.29 |
|    | HR | 4.37 ± 0.71 | 6.30 ± 1.08 | 3.90 ± 0.92 | 14.57 ± 2.50 | 1.37 ± 0.29 |
| T3 | DR | 4.39 ± 1.10 | 7.11 ± 1.82 | 3.74 ± 0.75 | 15.24 ± 3.47 | 1.24 ± 0.20 |
|    | PC | 4.38 ± 1.28 | 6.88 ± 1.73 | 4.12 ± 0.73 | 15.38 ± 3.35 | 1.34 ± 0.22 |
|    | HR | 4.17 ± 1.10 | 7.12 ± 1.46 | 3.84 ± 1.42 | 15.13 ± 3.48 | 1.21 ± 0.32 |
| T6 | DR | 4.21 ± 0.55 | 6.87 ± 1.38 | 3.84 ± 1.23 | 14.93 ± 2.95 | 1.20 ± 0.29 |
|    | PC | 5.03 ± 1.33 | 6.64 ± 1.22 | 4.08 ± 1.43 | 15.75 ± 3.78 | 1.42 ± 0.47 |
|    | HR | 4.09 ± 0.58 | 6.03 ± 1.05 | 3.12 ± 0.55 | 13.24 ± 1.27 | 1.09 ± 0.09 |

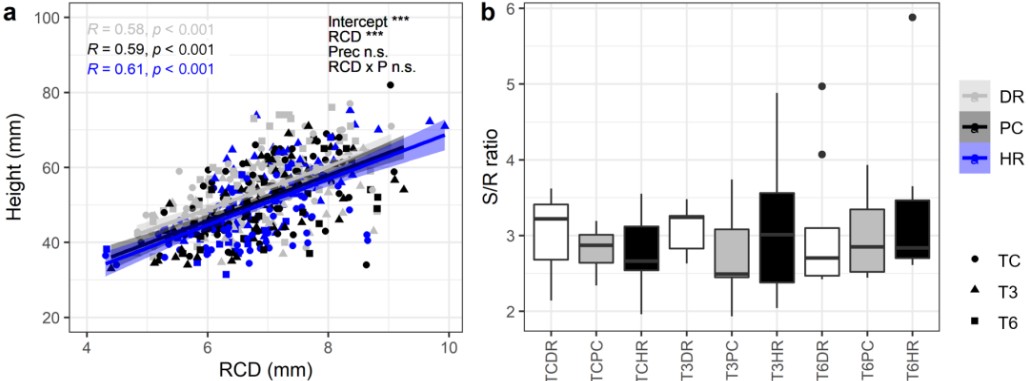

**Figure 5.** (**a**) Relationship between root collar diameter and height and (**b**) shoot to root ratios at the end of the treatment of *L. kaempferi* seedlings subjected to extreme climate events. TC: ambient temperature, T3: +3 °C warming, T6: +6 °C warming, DR: drought, PC: ambient precipitation, HR: heavy rainfall. Results from the ANOVA and Pearson correlation analysis are shown in panel (**a**). Asterisks indicate significant differences between warming treatments. Different letters indicate significant differences between treatments. The boxes' lower and upper limits indicate the 25% and 75% quartile, respectively. Bold lines within the boxes indicate the median, whiskers indicate the range of values from the minimum to maximum, and data points beyond the end of the whiskers are then plotted as filled dots.

## 4. Discussion

### 4.1. Survival

We hypothesized that extreme warming, drought, and heavy rainfall would decrease survival rates. In general, warming-induced heat stress can decrease tree growth and thus frequently decreases the survival rate [9]. In temperate regions, extensive drought can induce large-scale tree population declines [48–50]; however, some field studies reported that severe drought treatments did not significantly alter primary production [7]. Our study showed that the experimental climate events did not affect seedling survival at the end of the experimental period. In our open-field experiment, we used one-year-old container seedlings, which can be advantageous for drought avoidance potential, resulting in higher levels of field survival when compared to bare-root seedlings in terms of planting stress and early field performance [51,52]. As we applied the treatments in July, two months after the planting of the seedlings, the seedlings had sufficient time to stabilize and adapt to the field environments, compared to other experiments conducted during different periods including the early growing season [16,53]. Therefore, such manipulation experiments should be executed over multiple years, rather than within a single year, particularly with

respect to variation in air humidity [15]. Furthermore, long-term field experiments are required to determine the impact of drought on tree mortality because plants may show a time lag regarding the effects of drought on productivity and mortality [7,54].

### 4.2. Plant Growth

We expected that moderate warming would increase seedling growth in terms of RCD, height, and biomass, whereas extreme warming should reduce growth. In general, increasing temperatures stimulate plant biomass accumulation, but temperature-induced heat stress and water deficit reduce plant growth, thereby offsetting the stimulation effect [7,55–58]. The temperature treatments (T3 and T6) of the current study did not significantly affect the overall growth performance; however, seedling height in the HR treatments was significantly larger in T3HR than in TCHR and T6HR (Figure 4c), which showed consistent patterns regarding stem volume following the order T3HR (40.3% increase) > TCHR > T6HR (5.2% decrease; Table S1). These results suggest that extreme temperatures within a certain range of water availability decrease plant growth [9].

In our study, seedling height was highest under experimental warming by 3 °C and heavy rainfall treatment (Figure 4c), showing that optimum levels of warming without drought stress can stimulate plant growth. However, the seedling height in T3HR was not significantly higher than that in TCPC, which is the ambient condition. Numerous studies on experimental warming have reported that increasing the temperature stimulates plant growth [59,60], whereas extreme climate conditions typically reduce plant growth and primary production [6,61]. A meta-analysis showed that warming increased plant carbon pools by 6.8% and 7.0% regarding above- and belowground parts, respectively, with significantly higher responses in forests, especially in tree seedlings and saplings, than in other ecosystem types [60]. In general, warming by 3 °C and 6 °C increased emergence, development, and shoot length in 15 North American tree species, and the seedlings showed increased growth in response to warming, whereas exceedingly dry or wet conditions limited this positive response [9].

We also hypothesized that drought and heavy rainfall would decrease seedling growth performance. Precipitation manipulation did not affect the RCD of the seedlings; however, we found a significant precipitation effect on seedling height and a significant interaction effect of precipitation and temperature on seedling height and stem volume, which is a function of both shoot height and RCD. For instance, heavy rainfall decreased seedling height and stem volume under TC and T6 treatments, but increased height under the T3 treatment. Heavy rainfall results in water logging, which impedes gas exchange and limits oxygen availability, thus restricting plant photosynthesis and production [7,62]. Our previous study on *L. kaempferi* seedlings observed that a higher precipitation-induced increase by 32.5% in soil water content decreased the total chlorophyll content and net photosynthetic rate by 8.24% and 4.55%, respectively [63]. Another previous study reported that short-term waterlogging strongly induced stomatal closure and reduced net photosynthesis in *L. kaempferi* seedlings [18]. The irrigation system in our well-drained soil did not result in water logging; however, soil moisture in HR treatments reached 31 vol %. Manipulated amounts of rainfall produced significant differences in mean soil moisture content at the surface soil layer (5 cm soil depth) between treatments; however, we did not monitor soil moisture at deeper soil layers. Thus, such measurements in deeper layers including the rooting zone are required to validate the effects on soil moisture conditions, and additional experiments under drier conditions are needed to observe more conclusive effects regarding available water content and drought stress.

In the current study, drought did not affect seedling height in the TC and T3 treatments. The seedling height under the T6 treatment was greater in the DR, rather than reduced, in the PC and HR, but did not differ from that in the TCPC treatment (Figure 4c). A previous study on the effect of drought on *L. kaempferi* seedlings highlighted that lower precipitation increased the total chlorophyll content and net photosynthetic rate by 6.40% and 4.32%, respectively, indicating that this species can have positive physiological responses to mod-

erate drought [63]. In addition, higher accumulated precipitation during the experimental period (July to August 2020) may reduce water stress and heat stress in the seedlings. The number of precipitation days (28 days) and the accumulated precipitation (832 mm) during the experimental period was 30% and 20% higher, respectively, than the average values over the past 30 years [36]. Although tolerance to drought depends on species, soil type, and drought intensity and duration [27,64,65], our results suggest that short-term drought events in summer may not constitute extreme climatic events for *L. kaempferi* seedlings, whereas excessive rainfall during short periods may affect plant growth.

### 4.3. Biomass Allocation and Seedling Quality

In general, moderate warming enhances plant growth due to the increased allocation of carbon to aboveground rather than belowground parts, thereby increasing the shoot-to-root ratio or total biomass-to-root biomass ratio [66–68]. By contrast, drought-induced stress increases the root-to-shoot ratio by increasing the proportion of soluble sugars and starch in roots [16,69,70]. Thus, these climatic factors may exert interaction effects on carbon allocation and metabolic activity [67,71]. In our study, similar to the lack of remarkable responses of height growth to soil drought, no significant effect of drought on S/R ratios was observed (Figure 5b). The responses of carbon allocation to heat and drought stress can also vary according to species-specific water use strategy, season, and development stage [9,56,66,71–73]. Taeger et al. [56] showed that experimental drought increased taproot length and root-to-shoot ratios of *Pinus sylvestris* L. seedlings. Arend et al. [67] showed that the ratio of root length to shoot height increased in response to drought in *Quercus* species, but decreased in response to air warming, indicating provenance-specific sensitivity to drought and air warming. A previous study reported that an 80% reduction in precipitation led to a reduction in carbon isotopic composition for *L. kaempferi* seedlings, but did not affect stomatal conductance, predawn leaf water potential, and transpiration [17]. However, there has still been no comparable research in this species on the response of biomass allocation to warming and drought. In our current study, short-term drought stress in summer may not increase the root to shoot ratio for *L. kaempferi* seedlings as drought could alter the root structures by allocating more carbon to 1st- and 2nd-order fine roots than coarse roots [74]. Thus, further studies on drought responses to root morphological and physiological traits such as mean root diameter, specific root length, and respiration will be prerequisites to understanding biomass allocation for various species in a changing climate.

Regarding seedling quality at the nursery stage, we expected that extreme warming and heavy rainfall would reduce seedling quality. Practical guidelines for seedling management propose grading criteria for characterizing seedling quality, and nurseries commonly rely on morphological traits such as height and RCD as indicators of seedling quality [23,43]. According to the Korean practical nursery guidelines, the standard measurements of good-quality 1-1 *L. kaempferi* seedlings are 35–60 cm height, >6 mm RCD, and a H/D ratio <90 at a height exceeding 60 cm [37]. In our study, the measurements of a few seedlings per treatment were outside these ranges, but the mean values of all treatments were within the ranges of good-quality seedlings. Significant differences in the H/D ratio were mainly due to significant variations in height between treatments (Table S1). These results indicate that our short-term climate treatments altered the H/D ratio of 1-1 *L. kaempferi* seedlings, but they did not critically reduce seedling quality. The SQI, of which higher values are preferable, reflects changes in biomass, H/D, and S/R ratios, and thus appears to be a good index for describing seedling quality [23,45]. The SQI of germinated seedlings of *L. kaempferi* decreased as the temperature or precipitation increased [26]. In our study, there was no significant difference in SQI between the treatments (Table 2). This suggests that short-term extreme climate events during humid summers do not significantly affect the quality of *L. kaempferi* seedlings.



## 5. Conclusions

Overall, seedling biomass, biomass allocation and mortality were not significantly affected by climate manipulation, indicating tolerance to short-term extreme climate events in summer. By contrast, moderate warming with increased precipitation (e.g., T3HR) in well-drained soils may constitute suitable nursery conditions for *L. kaempferi* seedlings. However, more frequent heat stress and a high soil water content due to heavy rainfall reduce seedling quality, as evidenced by the reduced height, volume, and SQI in T6HR compared to other treatments. Our results highlight that, regarding viability and early growth, *L. kaempferi* seedlings may tolerate short-term extreme warming events in summer, supporting the idea that *L. kaempferi* is beneficial to reforestation and restoration in arid woodlands and in the context of climate change. However, to elucidate the potential impact on tree growth performance, further studies are needed to examine the effects of higher frequencies and longer periods of extreme climate events on various species regarding growth and other physiological responses.

**Supplementary Materials:** The following are available online at https://www.mdpi.com/article/10.3390/f12111595/s1, Table S1: Survival rate, root collar diameter, height, stem volume, H/D, S/R, and T/R ratios of *Larix kaempferi* seedlings under extreme climate events by temperature and precipitation manipulations. TC: ambient temperature, T3: +3 °C warming, T6: +6 °C extreme warming, DR: drought, PC: ambient precipitation, HR: heavy rainfall. Values are means ± S.D. ($n = 3$ for survival rate, $n = 90$ for RCD, height, stem volume and H/D ratio, $n = 9$ for S/R and T/R ratios. Values with different letters in a column indicate statistical differences between treatments within a species at 5% levels by Tukey HSD test, Figure S1: The descriptions of (**a**) warming treatment system and (**b**) areas warmed by infrared heating lamps and locations of the sampled seedlings (30 samples for root collar diameter and height growth measurements and three samples for biomass measurement. (modified from Kim et al., under review), Figure S2: The relationship between stem volume and stem biomass of *Larix kaempferi* seedlings across the temperature and precipitation manipulations. TC: ambient temperature, T3: +3 °C warming, T6: +6 °C extreme warming, DR: drought, PC: ambient precipitation, HR: heavy rainfall. Values are means ± S.D. ($n = 9$ for stem biomass, $n = 60$ or $90$ for stem volume per treatment). Results from the Pearson correlation analysis are shown in the figure.

**Author Contributions:** Conceptualization, N.-J.N. and Y.S.; methodology, N.-J.N. and Y.S.; formal analysis, N.-J.N.; investigation, N.-J.N., G.-J.K., and M.-S.C.; resources, N.-J.N.; data curation, N.-J.N.; writing—original draft preparation, N.-J.N.; writing—review and editing, N.-J.N., G.-J.K., Y.S., and M.-S.C.; visualization, N.-J.N.; supervision, M.-S.C.; project administration, Y.S. and M.-S.C.; funding acquisition, Y.S. All authors have read and agreed to the published version of the manuscript.

**Funding:** This research was funded by the Korea Forest Service, grant number 2020181A00-2122-BB01.

**Institutional Review Board Statement:** Not applicable.

**Informed Consent Statement:** Not applicable.

**Data Availability Statement:** Not applicable.

**Acknowledgments:** This work was carried out at the Forest Technology and Management Research Center. We thank Won-Geuk Kim for nursery settings for the experiment.

**Conflicts of Interest:** The authors declare no conflict of interest.

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
