# Peer review of "Early Growth Responses of Larix kaempferi (Lamb.) Carr. Seedling to Short-Term Extreme Climate Events in Summer"

_forests, doi:10.3390/f12111595_

Round 1
Reviewer 1 Report
Please, in the attached file is the review report.
Author Response
- GENERAL COMMENTS
I consider that the topic of the manuscript is interesting and bring new information about the responses of Larix k. to climate change events (drought, extreme precipitations, temperature rise). For this, this study can serve to understand the management of this species in reforestation planning, and their natural regeneration in the context of climate change.
The experimental design is in general novel and can also serve as a guide for other similar studies in other species, and at worldwide level. The authors have done a very good work measuring seedlings.
However, the manuscript is missing important points of a standard publication and needs extra improvement in order to provide a better view of the studied topic. My major concerns are regarding the data analysis, and the discussion section. The authors should clarify the experimental design and their analysis. The trial period has been short (only a few months); therefore authors must i) emphasize the most important results, and ii) better discuss these main results. Few variables have been significantly affected by the treatments. For this, the discussion should be summarized, and focusing on what is most important. I see many weaknesses of the text that need major revision before acceptance. Therefore, I have made a list of comments that I would like to be widely reviewed by the authors.
General Response: We appreciate your valuable comments. We agree that the experimental period is short, thus we emphasized the important results and better discussed the main results as the reviewer commented. We also shortened the discussion section, and focused on drought tolerant responses and heavy rainfall vulnerable responses following your specific comments to improve our manuscript.
- SPECIFIC COMMENTS
- Title
Comment 1: Larix kaempferi (Lamb.) Carr.
Response 1: Thanks for your comments. We corrected it.
- Abstract
Comment 2: Line 17. ºC
Response 2: We corrected it.
Comment 3: Pag. 19: change “estimated” by “measured”
Response 3: We changed “estimated” by “measured”.
Comment 4: Line 20: “seedling growth” is too vague, include height growth, biomass growth..
Response 4: Thanks for your comment. We clarified it by including “height growth”, as you commented.
Comment 5: Line 22: “seedling quality” is too vague; include “seedling quality index”.
Response 5: Thanks for your comment. We corrected it as you commented.
Comment 6: It is necessary to include the more important, numerical results (not only a description of results)
Response 6: Thanks for your comments. We include more important numerical results such as survival rate and percentage decreases in stem volume under heavy rainfall treatment. Please check the abstract.
Line 18-22: “The survival rate of seedlings did not differ among treatments, showing high values exceeding 94% across treatments. The measured shoot height was largest under warming by 3 °C and high rainfall, indicating that moderate warming increased seedling height and biomass growths in a moist environment. Heavy rainfall decreased stem volume by 21% and 25% under control and warming by 6 °C treatments, respectively”
- Introduction
Comment 7: Line 41: “excessive rainfall may decrease forest productivity”; I think this is discussible, because this phenomenon can only occur under an extreme rainfall, and in short periods of temp.
Response 7: Thanks for your comments. We agree with you comment. We described it more specifically by adding “under extreme rainfall”. Please check the line 52-54.
Line 52-54: “Moreover, excessive rainfall may also decrease plant productivity under extreme rainfall due to reduced photosynthetic and stomatal conductance under waterlogged conditions [18].”
Comment 8: Line 59: Larix kaempferi (Lamb.) Carr.
Response 8: Thanks for your comments. We added the binomial name as you commented.
Comment 9: Line 61. Delete “in Korea”. Perhaps Asia is a more widespread distribution
Response 9: Thanks for your comment. We changed “Korea” by “Asia”.
Comment 10: Line 61-64. Summarize this paragraph, indicating the geographic area of the data
Response 10: Thanks for your comment. The paragraph has been summarized, indicating “across South Korea” because the planting areas are distributed across South Korea. Please check the line 73-76.
Line 73-76: “The plantation areas of L. kaempferi gradually increased approximately ten-fold over the past decade across South Korea, and the planted area in 2019 accounted for approximately 20% (4,559 ha) of the total plantation area”
Comment 11: Line 68: authors included in this paragraph the species L. principis-rupprechtii. Please, modify this paragraph, for example: “…A few Previous studies suggested a high risk of growth cessation in similar Larix species such as L. principis-rupprechtii planted across large areas in north-central China during extreme drought”.
Response 12: Thanks for your comment. The sentence has been revised as you commented. Please find the line 68-70.
Line 80-82: “A few previous studies suggested a high risk of growth cessation in similar larch species such as L. principis-rupprechtii planted across large areas in north-central China during extreme drought”
Comment 12: Line 78. Reference (26). Warming 6 ºC
Response 12: Thank you, but the warming temperature in the reference was 3 ºC is correct.
Comment 13: Line 80, objectives. “Therefore, the objective of this study was to determine the growth performance of larch seedlings under experimental warming and precipitation conditions”. Seedling survival is another aim of this study, please incorporate it
Response 13: Thanks for your comment. We incorporated it to the objective. Please find the line 94-96.
“Therefore, the objective of this study was to determine the survival rate and growth performance of larch seedlings under experimental warming and precipitation conditions.”
Comment 14: Line 85: “seedling quality” is too vague, and it is and indicator in function of other variables. Include the main variables
Response 14: We agree with your comment. We revised the sentence. Please check the line 94.
Comment 15: In the introduction section, it is recommendable include more information about the ecological requirements and temperament of Larix to growth.
Response 15: Thanks for your comment. We include more information about the larch species.
Line 76-78: “Larix kaempferi is an important fast- growing deciduous coniferous species encircling Northern Hemisphere, due to disease- and cold-resistances compared to other planting species [17,30-34].”
- Methods
Comment 16: Line 92: indicate separation between seedlings into the plots
Response 16: Thanks for your comment. We planted seedlings with 11-cm intervals between seedlings. In addition, we provided a supplementary figure for readers to easily understand the processes as the reviewer 2 commented. Please check the line 108 and Figure S1b.
Figure S2. The descriptions of (left) warming treatment system and (right) areas warmed by infrared heating lamps and locations of the sampled seedlings (30 samples for root collar diameter and height growth measurements and 3 samples for biomass measurement. (modified from Kim et al., under review)
Comment 17: Figure 2. This is a good figure. Perhaps, it is necessary to change “Box” by “Block” in accordance with the experimental design
Response 17: Thanks for your comments. We changed ‘box’ by ‘block’ as you commented.
Comment 18: Line 126: “moisture content”, is “soil water content”. Please indicate the units
Response 18: Thanks for your comment. We changed ‘moisture content’ by ‘volumetric soil moisture content (vol %)’ with the unit.
Comment 19: Line 126: soil temperature (ºC)
Response 19: We added the unit.
Comment 20: Line 132. Variables should be design by acronyms (for example, height is H)
Response 20: Thanks for your comment. We added H, which is the acronyms of height.
Comment 21: Line 136: In October, three of the surviving seedlings per block were harvested to measure biomass accumulation and allocation (n = 9 per treatment).
Response 21: We added ‘per block’ as you commented.
Comment 22: Line 140. Stem volume: indicate units
Response 22: We indicated the unit (cm3).
Comment 23: Line 141, equation 1. Include acronyms in equations (Height = H)
Response 23: Thanks for your comment. We used the acronyms.
Comment 24: Line 142. Specify acronyms for root (R) and shoot biomass (S)
Response 24: We specified the acronyms.
Comment 25: Equation 2: Include the variables with their acronyms; for example, seedling total biomass is “T”, and after, specify the acronym
Response 25: Thanks for your comment. We indicated the variables with their acronyms and then used the acronyms for the following equation (2).
Comment 26: Line 152. “To determine the effect of treatments on tree growth with the exclusion of the influence of initial seedling size, a linear mixed-effect model was fitted using the ‘lme’ function”. I don't understand the meaning of “tree growth”. It refers to “height growth”, “diameter of root growth”, “biomass growth”…or “the growth of all the variables”. It is necessary to clarify this definition and refer it to the morphological variables
Response 26: Thanks for your comment. Tree growth should be height growth because we double checked the effect of precipitation treatment on height growth based on the results from two-way ANOVA. We clarified it as “height” growth.
Comment 27: Line 155. “One treatment was considered a fixed factor, the other treatment was a random factor, and the initial RCD was a covariate”. There are 9 treatments in the experimental design; why do the authors indicate that “one treatment” is “a fixed factor” and the other random? The explanation of this data analysis is very confusing and, perhaps, incorrect
Response 27: We agree that the explanation was very confusing. Yes, there were 9 treatments in the experimental design, but those are two factors. We double checked the effect of precipitation on the height growth with a random factor of temperature and a covariate of initial RCD. When considered the results from two-way ANOVA, temperature was not a significant factor affecting the growths. We, however, also tested the growth responses by using temperature as a fixed factor and precipitation as a random factor, and the initial RCD as a covariate. However, because we only showed the effect of precipitation on the height growth with a covariate of initial RCD in Figure 5a, we clarified the explanation. Please find the line 180-184.
- Line 180-184: “To determine the effect of precipitation treatment on tree height growth with the exclusion of the influence of RCD, a linear mixed-effect model was fitted using the ‘lme’ function of the ‘nlme’ R package [46]. The precipitation treatment was considered a fixed factor, the seedlings within blocks was a random factor, and the RCD was a covariate.”
Comment 28: The authors have made a linear mixed-effect model; they should indicate the model
Response 28: The lme model had been made by R library. I have already explained what the factors and covariate are. I don’t think we need to indicate additional description. Please check the script that I ran in R as follows.
### Linear mixed effect model
### Precipitation is fixed factor with a covariate of RCD, temperature is random factor
> extraw90 <- read.csv("2020extreme_eng_n90_filtered.csv")
> library(lme4)
> library(car)
> model.lmer <- lmer(H ~ RCD*Ptreat + (1|block) + (1|seedling), data=extraw90.L), data=extraw90)
> summary(model.lmer)
Linear mixed model fit by REML ['lmerMod']
Formula: H10 ~ D10 * Ptreat2 + (1 | rep) + (1 | no)
Data: extraw90.L
REML criterion at convergence: 3639.3
Scaled residuals:
Min 1Q Median 3Q Max
-3.7938 -0.6623 0.0012 0.6957 3.5862
Random effects:
Groups Name Variance Std.Dev.
no (Intercept) 0.6924 0.8321
rep (Intercept) 7.2506 2.6927
Residual 52.4812 7.2444
Number of obs: 535, groups: no, 30; rep, 3
Fixed effects:
Estimate Std. Error t value
(Intercept) 20.0025 4.1820 4.783
D10 4.9775 0.5635 8.834
Ptreat2P1 -11.5166 5.7168 -2.015
Ptreat2P2 -8.7590 5.7648 -1.519
D10:Ptreat2P1 1.1426 0.8246 1.386
D10:Ptreat2P2 0.7476 0.8397 0.890
Correlation of Fixed Effects:
(Intr) D10 Ptr2P1 Ptr2P2 D10:P2P1
D10 -0.919
Ptreat2P1 -0.640 0.684
Ptreat2P2 -0.607 0.651 0.453
D10:Ptrt2P1 0.636 -0.692 -0.991 -0.453
D10:Ptrt2P2 0.603 -0.658 -0.450 -0.991 0.458
> Anova(model.lmer, type=3)
Analysis of Deviance Table (Type III Wald chisquare tests)
Response: H10
Chisq Df Pr(>Chisq)
(Intercept) 22.8774 1 1.727e-06 ***
D10 78.0320 1 < 2.2e-16 ***
Ptreat2 4.5212 2 0.1043
D10:Ptreat2 2.0031 2 0.3673
---
Signif. codes: 0 ‘***’ 0.001 ‘**’ 0.01 ‘*’ 0.05 ‘.’ 0.1 ‘ ’ 1
>
Comment 29: Indicate the statistical software used for the ANOVA (R?), and include the model. Analyze the inclusion of the 3 blocks in the experimental design
Response 29: Yes, we used the R software for all statistics including the ANOVA, Tukey test and lmer, and making all figures. We indicated it. Regarding to the model, we answered to your comment above. It’s a shame that we could not include the 3 blocks design in our analysis because one replicate of T6PC and T6HR in a block was missed. We explained it. Please check line 175-179 and 186-187.
Line 175-179: “We excluded one replicate from the T6PC and T6HR treatments as two plots among the 27 randomized complete block-designed plots were partly malfunctional. Consequently, we analyzed the data from each treatment (n = 60 or 90 per treatment for RCD and height, n = 9 per treatment for biomass).”
Line 186-187: “All statistical analyses were performed with the available variables measured in May and October using R version 4.1.1 [47].”
Comment 30: Was the ANOVA performed with the variables measured in May or in October? The period or months, Is it a variable?
Response 30: Yes, the ANOVA performed with the variables measured was in both May and October. However, the month was not a variable. We have shown the results in supplementary Table S1 to confirm that there was no significant difference in variables measured in May among treatments (pre-treatment). To clarify it, we added two sentences in the statistics and result sections. Please check the line 186-187
Line 186-187: “All statistical analyses were performed with the available variables measured in May and October using R version 4.1.1 [47].”
Line 210-212: “There was no significant difference in the RCD and height measured in mid-May among pretreatments (Table S1).”
Comment 31: Authors should revise and clarify all the data analysis
Response 31: Thanks for your comments. We revised and clarified all the data analysis in the section.
- Results
Comment 32: 3.1.1. “Temperature and moisture” section. Change by soil temperature and soil water content, indicating units. This section should be “experimental conditions of soil”. Or perhaps, indicate “effects of temperature and precipitation treatments on seedling development” (see below)
Response 32: Thanks for your comment. We changed the section title by “Effects of temperature and precipitation treatments on seedling development” and deleted the sub-sections as you commented.
Comment 33: Line 161. Warming treatment is W?. Please specify. Authors should use acronyms in all the manuscript
Response 33: We prefer to use temperature treatment including TC, T3, and T6 rather than warming treatment because ambient temperature is a level of the treatments.
Comment 34: Table 1. The block effect is missing in the ANOVA table. In the experimental design the “Block” is another main factor. It is desirable that the “block” is not significant, but it should be included with 2 degrees of freedom in the ANOVA. Why haven't the authors included the “block” in the model?
Response 34: We could not include the 3 blocks design in our analysis because one replicate of T6PC and T6HR in a block was missed, as we responded to above comments. Nevertheless, we could confirm that block was not significant when we tested for available data excluding T6, for example TC vs T3 effects on the variables. We explained it. Please check the answer to comment 30.
Comment 35: Table 1. H/D ratio is missing
Response 35: Thanks for your comment. We added H/D into the Table 1. Please check the Table 1.
Comment 36: Table 1. Total mass, it is total biomass (g)? These variables are also analyzed in Table 2
Response 36: Thanks for your comment. Yes, total mass is total biomass. We corrected it in the Table 1. Table 1 showed the statistical results from ANOVA for the variables, while Table 2 showed the mean values with no significant difference in the measurements among 9 treatments. I think there is no issue to show the statistical results in Table 1 and the means in the Table 2.
Comment 37: Table of ANOVA. The indicated factors are “temperature” and “precipitation”, whereas authors indicated “warming treatment” in the text. Please, homogenize the factors
Response 37: Thanks for your comment. We homogenized the factor as “temperature treatment”.
Comment 38: Figure 3. Figure 3 should be inserted before table 1 (experimental conditions)
Response 38: Thanks for your comment. We inserted Figure 3 before Table 1.
Comment 39: Figure 3 caption. “Different letters indicate significant differences among treatment within each group. Factor”
Response 39: Thanks for your comment. We added ‘factor’ to the caption.
Comment 40: Figure 3. Y axis. It is soil temperature
Response 40: We corrected it. Also, we changed “soil moisture” by “soil water content”.
Comment 41: It is necessary to include a section indicating the performed ANOVA. This section is confusing
Response 41: Thanks for your comment. We revised the section to clearly indicate the performed ANOVA on the variables by showing significant level (p < 0001) at the first part of each paragraph. Please check them in the Result section.
Comment 42: In Table 1, “environment” is the experimental condition of soil
Response 42: We changed “environment” by “experimental soil condition”.
Comment 43: Section 3.1.2. This section is not necessary because a previous section describing the factor effects was included in the manuscript.
Response 43: As you commented, we deleted the subsection.
Comment 44: Line 184. The height of seedlings was largest in T3HR (55.6 mm) and smallest in TCHR and T6HR (45.6 mm; p < 0.05, Figure 4c). I think this is the main result of the paper, and authors should emphasize it. The high temperature and extreme rainfall increased stem height; but what is this effect due to?
Response 44: We wanted to show the maximum and minimum values of seedlings height. Main result was significantly lower height in TCHR and T6HR compared to that in TCPC. The moderate warming and extreme rainfall (T3HR) did not significantly increase stem height when compared to that in ambient temperature and precipitation condition. There was no difference in mean soil water content among TCHR, T3HR and T6HR. However, we assumed that soil water contents could be too wet in TCHR and repeated heat and moisture stress in T6HR could decrease height. Our future research will include measuring physiological traits under extreme events. the reason. Thanks for your insightful comment.
Comment 45: Figure 4c. The treatment T6DR appears as significant. Why haven't the authors indicated it?
Response 45: T6DR appears as significant within the T6 group, but drought did not increase seedling height growth compared to ambient temperature and precipitation condition, showing the seedling would be tolerant to water deficit. We revised and indicated the reason in the discussion section 4.2. Please find the line 325-335.
Line 325-335: In the current study, drought did not affect seedling height in the TC and T3 treatments. On the other hand, the seedling height under the T6 treatment was greater in the DR, rather than reduced, in the PC and HR, however did not differ from that in the TCPC treatment (Figure 4c). A previous study on the effect of drought on L. kaempferi seedlings highlighted that lower precipitation increased total chlorophyll content and net photosynthetic rate by 6.40% and 4.32%, respectively, indicating that this species can have positive physiological responses to moderate drought [63]. In addition, higher accumulated precipitation during the experimental period (July to August 2020) may reduce water stress and heat stress in the seedlings. The number of precipitation days (28 days) and the accumulated precipitation (832 mm) during the experimental period was 30% and 20% higher, respectively, than the average values over the past 30 years [36].
Comment 46: Section 3.1.3. Again, this section is not necessary because a previous section describing the factor effects was included in the manuscript. The treatments that affect height and stem volume are the same.
Response 46: We deleted the sub-section as you commented.
Comment 47: Table 2. The variables are “leaf biomass”, “shoot biomass”…”total biomass”
Response 47: Thanks for your comment. We revised them, as you commented.
Comment 48: Stem volume and stem biomass are directly correlated. I do not understand that the treatments (interaction) affect the stem volume but not the shoot biomass. The authors should discuss whether this is because treatments do not affect the diameter. Please, explain this question
Response 48: Thanks for your comment and question. Stem volume are correlated with stem biomass, but it doesn’t mean that the relationship accounts for 100%. As you know, those are not the same growth parameters. It also makes a sense that differences in sample numbers could result in differences in the statistic results. We do not think that it is necessary to discuss it. Instead, we just briefly explained the reason why in the result section and we added the relationship to a supplementary Figure S1. Please check the line 231-234.
Line 231-234: “The small sample size may result in no significant difference in biomass accumulation among the treatments, however stem biomass was directly correlated with stem volume across the treatments (p < 0.001, Figure S2).”
Supplementary Figure S2. The relationship between stem volume and stem biomass of Larix kaempferi seedlings across the temperature and precipitation manipulations. TC: ambient temperature, T3: +3 °C warming, T6: +6 °C extreme warming, DR: drought, PC: ambient precipitation, HR: heavy rainfall. Values are means ± S.D. (n=9 for stem biomass, n=60 or 90 for stem volume per treatment). Result from Pearson correlation analysis is shown in the figure.
Comment 49: Line 204. The correlation analysis (regression) between height and RCD are not included in the methodological section; why?
Response 49: Thanks for your comment. The correlation between height and RCD was analyzed by functions of ‘stat_cor’ of ‘ggpubr’ R package. We added the explanation in the methodological section. Please check the line 184-186.
Line 184-186: “Pearson correlations between height and RCD were analyzed using the ‘stat_corr‘ function of the ‘ggpubr’ R package.”
- Discussion
Comment 50: Discussion is large and should be shortened. especially for survival. Survival analysis for a few months is a very short time to obtain significant results. I believe that the authors cannot discuss much about this variable
Response 50: Thanks for your comment. We agree with you comment. We have shortened the section by deleting several discussion lines. Please find the revised section 4.1.
Comment 51: Line 255. Discuss the effects of the treatment T6DR (figure 4c)
Response 51: Thanks for your comment. We discuss the effects of the treatment T6DR. Please check the response to your comment 45 above.
Comment 52: Line 258. Seedling biomass (root, stem, leaf, and total biomass) did not differ among treatments. In consequence, treatments for this variable cannot be compared or related to those obtained for stem volume. Authors should delete these statements related to biomass analysis
Response 52: We deleted the statements related to biomass analysis as you commented.
Comment 53: Line 261. But in this study is the contrary (3 ºC stimulated height growth). Why? This seems like a contradiction
Response 53: Thanks for your question and comment. There was a different experimental setting between the two studies. Numerous studies reported that moderate warming increases height growth for many species. However, the previous study showed decrease in the physiological traits as the experiment manipulated the 3 ºC warming during the whole growing season for two year, which was drier than in current study year. The prolonged treatment may significantly decrease the physiological traits. We discussed this point in the following paragraphs and other parts in the discussion section. We also deleted the statement because the study introduced in the introduction section to build a hypothesis. Please check the line 292-294.
Line 292-294: “In our study, seedling height was significantly highest under the experimental warming by 3 °C and the heavy rainfall treatment (Figure 4c), showing that optimum levels of warming without drought stress can stimulate plant growth. However, the seedling height in T3HR was not significantly higher than that in TCPC which is ambient condition.”
Comment 54: Line 271. The authors refer to a semi-arid climate. It probably cannot be compared with the treatments performed. Authors confirmed this by including the reference 8
Response 54: Thanks for your comment. We deleted the comparison.
Comment 55: Lines 299-302. Interesting results. For this, could this species be indicated for forest restoration in arid woodlands and in the context of climate change? This question can be included in the conclusions section
Response 55: Thanks for your comment. We are careful to determine if this larch species would be beneficial to forest restoration or reforestation in arid woodlands in the context of climate change based on our short-term study. However, as you commented, we included this question in the conclusion section. Please check the line 390-393.
Line 390-393: “Our results highlight that, regarding viability and early growth, L. kaempferi seedlings may tolerate short-term extreme warming events in summer, supporting the possibilities that L. kaempferi is beneficial to reforestation and restoration in arid woodlands and in the context of climate change.”
Comment 56: 4.3. Biomass allocation and seedling quality. In general, drought significantly affect the shoot/root ratio in conifers, because plants allocate more C in roots. I think this issue is not sufficiently explained by the authors. The authors should explain this issue in terms of both the physiology and temperament of this species, and its responses to drought. Undoubtedly, the study period is short, and this is likely to have a decisive influence in the results.
Response 56: Thanks for your comment. We totally agree with your comment. Most studies on this species are on morphological and molecular, genomic and phylogenetic, progeny testing, breeding for disease resistance, wood quality perspective, and nutrient responses. We have a previous study on physiological response to drought. We added the empirical physiological traits in the discussion section. However, carbon allocation response of this specific species to drought has been poorly understood. That’s why this study has an originality to be published even though the study period is relatively short. Anyway, there could be a possible reason why drought does not increase root to shoot ratio. A short-term drought stress could not increase R/S ratio of young seedlings, but may alter root structure by allocating more carbon to 1st- and 2nd-order fine roots than coarse roots. We added some relevant discussion in the section. Please find the line xxx-xxx and xxx-xxx.
Line 313-315: “Our previous study observed that higher precipitation manipulation decreased the total chlorophyll content and net photosynthetic rate by 8.24% and 4.55%, respectively [63].”
Line 328-331: “A previous study on the effect of drought on L. kaempferi seedlings highlighted that lower precipitation increased the total chlorophyll content and net photosynthetic rate by 6.40% and 4.32%, respectively, indicating that this species can have positive physiological responses to moderate drought [63].”
Line 354-363: “A previous study reported that 80% reduction of precipitation showed a reduction tendency in carbon isotopic composition for L. kaempferi seedling, but did not affect stomatal conductance, predawn, leaf water potential and transpiration [17]. However, there was still no comparable research on this species in the response of biomass allocation to warming and drought. In our current study, short-term drought stress could not increase R/S ratio of young seedlings, but may alter root structure by allocating more carbon to lst- and 2nd-order fine roots than coarse roots [73]. Thus, further studies on drought responses to root morphological and physiological traits such as mean root diameter, specific root length and respiration will be prerequisite to understand biomass allocation for various species under a changing climate.”
Comment 57: Line 332. “In our study, SQI was relatively low in T6HR (1.09) compared to other treatments (1.20 to 1.42), but we found no significant difference in SQI among the treatments (Table 2). This result suggests that short-term extreme climate events during humid summers do not significantly affect the quality of L. kaempferi seedlings”. The authors should not discus about this index because there were no significant differences between the treatments.
Response 57: Thanks for your comment. We deleted the comparison.
Comment 58: In general authors should better explain why shoot height increase in function of temperature, but not other variables related to shoot (biomass, and ratios)
Response 58: We had explained partly about biomass and R/S ratio as you commented. Please find the previous responses to your comments above. But it was quite difficult to explain the species-specific responses of physiological functioning in this study.
Comment 59: The stem volume should be emphasized in the discussion. This variable was also influenced by the treatments. The volume is a function of the shoot height and also the root collar diameter.
Response 59: Thanks for your comment. We included stem volume which is a function of both shoot height and root collar diameter to the discussion section. Please check the line 306-311.
Line 306-311: “Precipitation manipulation did not affect the RCD of the seedlings, however, we found a significant precipitation effect on seedling height and a significant interaction effect of precipitation and temperature on seedling height and stem volume which is a function of both stem height and RCD. For instance, heavy rainfall decreased seedling height and stem volume under TC and T6 treatments, but increased height under the T3 treatment.”
- Conclusions
Comment 60: Conclusion should be shortened (this section contains repetitions of the results and discussion)
Response 60: Thanks for your comment. We shortened our conclusion by deleting some repetitions of the results.
Comment 61: Include the possibilities of using this species to reforestation of arid woodlands and in the context of climate change
Response 61: Thanks for your comment. We included your good suggestion as we responded to your comment 55. Please check the line 390-393 in the discussion section.

Reviewer 2 Report
Dear Authors
The publication "Early Growth Responses of Larix kaempferi Seedlings to Short term Extreme Climate Events in Summer" discusses important issues of tree response to climate change for science and forestry. In the context of the article submitted for review, the research presented therein focused on Larix kaempferi and observed the response of seedlings to the occurrence of short-term extreme climate events.
In the review, I would like to highlight the high quality of the experimental setup, which requires considerable financial resources and the measurement equipment used. However, there are general questions about the model as well as about the conduct of the measurements themselves.
In Section 2.1, the authors describe an experimental design in which they raised the temperature by 3 and 6 degrees Celsius in a controlled manner, while the rainfall was 113 mm per day. The values were chosen based on the reference period (1961 - 2019). The title of the paper suggests possible future climate events, while references to hypothetical climate change models developed for the study region are missing in both the text and the proposed model. I have doubts about the choice of temperature and precipitation in the presented model. I would like you to describe in the introduction the theoretical models of climate change for the region and how they relate to the values chosen by the authors.
In the experiment, 88 seedlings were planted in each plot with a homogeneous sandy loam soil. The reviewer did not read anything in the text about the origin of the seeds. The authors completely omitted evolution and its mechanisms. Moreover, we as readers of the publication cannot deduce anything about the epigenetic mechanism influencing the observed processes. There are many publications describing different responses of the origin or progeny of single individuals to climatic variables. I consider the lack of data on seed origin to be a significant omission.
One of the most important questions to address is the number of samples measured. With 88 plants planted in each plot and high survival rates, why were only 30 plants measured. This represents only 0.34 of the experiment - in this reviewer's opinion, a considerable amount of describable variability was overlooked. There is no description of how the cuttings were selected for measurement (line 134 states where they were selected). Statistical tests require random selection, which to my knowledge was not followed. I would also note that only 3 plants in each plot were selected for detailed measurements (lines 136-137), the survival rate of the experiment would have allowed for a much larger sample. It is not obvious to me why more detailed measurements were not made, such as scanning the root systems and analysing them with common and available scientific analysis tools. If the authors believe that the sample selection was correct and that randomization was maintained, please explain and describe this in detail.
Please answer the questions that have been asked. I believe that a major revision of the publication is pending, and if it is accepted for publication, there will be further detailed revisions of the text.
Best regards
Reviewer
Author Response
General Comment: The publication "Early Growth Responses of Larix kaempferi Seedlings to Short term Extreme Climate Events in Summer" discusses important issues of tree response to climate change for science and forestry. In the context of the article submitted for review, the research presented therein focused on Larix kaempferi and observed the response of seedlings to the occurrence of short-term extreme climate events. In the review, I would like to highlight the high quality of the experimental setup, which requires considerable financial resources and the measurement equipment used. However, there are general questions about the model as well as about the conduct of the measurements themselves.
Comment 1: In Section 2.1, the authors describe an experimental design in which they raised the temperature by 3 and 6 degrees Celsius in a controlled manner, while the rainfall was 113 mm per day. The values were chosen based on the reference period (1961 - 2019). The title of the paper suggests possible future climate events, while references to hypothetical climate change models developed for the study region are missing in both the text and the proposed model. I have doubts about the choice of temperature and precipitation in the presented model. I would like you to describe in the introduction the theoretical models of climate change for the region and how they relate to the values chosen by the authors.
Response 1: Thanks for your good comment. We agree with your comment. Regarding definition of extreme events, an extreme climatic event has been defined in different ways which are based on statistical quantification of climatic variables and a synthetic definition in both driving and responses variables, and the tails of a distribution for a climate parameter climatologically. As you mentioned, there are possible future climate events based on climate change scenarios. One common way is based on statistical quantification of climatic variables. For example, the IPCC defines an extreme climatic event as one event being rare than the 10th or 90th percentile of climate events within its statistical frequency distribution at a particular place over a certain period of time. As you commented we described the model we chose in the introduction section. Please find the line xx-xx.
Comment 2: In the experiment, 88 seedlings were planted in each plot with a homogeneous sandy loam soil. The reviewer did not read anything in the text about the origin of the seeds. The authors completely omitted evolution and its mechanisms. Moreover, we as readers of the publication cannot deduce anything about the epigenetic mechanism influencing the observed processes. There are many publications describing different responses of the origin or progeny of single individuals to climatic variables. I consider the lack of data on seed origin to be a significant omission.
Response 2: Thanks for your comment. We agree that the origin of the seeds is very important information. We obtained seeds from a national seed orchard, Anmyeon-do, Chungnam, managed by National Forest Seed Variety Center. The center is under the Korea Forest Service. Then we produced the seedlings in an experimental tree nursery nearby our experimental site located in Forest Technology and Management Research Center. We described the details about the seed origin in the Material section. Please check the line 110-113.
Line 110-113: “The seeds obtained from a seed orchard located in Anmyeondo (36° 29’ N, 126° 23’ E, 40–50 m a.s.l.) of National Forest Seed Variety Center were grown at the container tree nursery in Forest Technology and Management Research Center. The annual mean air temperature and precipitation in the seed orchard are 13 °C and 1,380 mm, respectively [36].”
Comment 3: One of the most important questions to address is the number of samples measured. (1) With 88 plants planted in each plot and high survival rates, why were only 30 plants measured. This represents only 0.34 of the experiment - in this reviewer's opinion, a considerable amount of describable variability was overlooked. There is no description of how the cuttings were selected for measurement (line 134 states where they were selected). Statistical tests require random selection, which to my knowledge was not followed. (2) I would also note that only 3 plants in each plot were selected for detailed measurements (lines 136-137), the survival rate of the experiment would have allowed for a much larger sample. (3) It is not obvious to me why more detailed measurements were not made, such as scanning the root systems and analysing them with common and available scientific analysis tools. If the authors believe that the sample selection was correct and that randomization was maintained, please explain and describe this in detail.
Response 3(1): Thank you very much for your comments. Warming was controlled by 3 infrared heating lamps and infrared temperature sensors per each plot (Figure S2). As we considered well-warming treated seedlings, we measured 30 seedlings from near center of each plot and sampled 3 seedlings at center of each plot. Although we also confirmed that all planted seedlings were well-warmed by our warming systems by using IR camera, we wanted to select seedlings that theologically completely warmed by 3°C and°6 C in T3 and T6 treatment plots. We described the details about the sampling processes in the method section and provided a supplementary figure S1. Please find the line 158-161 and Figure S1b as below.
Line 158-161: “For growth measurements, 30 of 88 seedlings per plot were selected from near the center (n = 90 per treatment) as the 30 seedlings were relatively well-warmed by the infrared heating lamps when compared to the outer seedlings (Figure S1).”
Figure S2. The descriptions of (left) warming treatment system and (right) areas warmed by infrared heating lamps and locations of the sampled seedlings (30 samples for root collar diameter and height growth measurements and 3 samples for biomass measurement. (modified from Kim et al., under review)
Response 3(2): We agree that we could take a much larger samples for better analyzing biomass accumulation of seedlings. However, as we planned to continue to measure the variables for another year at that moment, we had to harvest a minimum number of samples. The number of harvested samples (n=3 per each plot, n=9 per treatment) was quite small, but it was all we could do. We harvested 3 seedlings per each plot at the center of each plot (Figure S2b). We made a figure to easily show how the samples were selected. As you commented, there was no description how the cuttings/harvestings were selected in the manuscript. We made the descriptions in the method section. Please check the line 162-164.
Line 162-164: “In October, three of the surviving seedlings per plot, grown from near the center of the plot, were harvested to measure biomass accumulation and allocation (n = 9 per treatment).”
Response 3(3): Regarding to your third comment, we again totally agree that we could make more detailed measurements in particular root system using some tools such as WinRhizo and some chemical properties such as nutrients in roots, but unfortunately, we did not measure them at that moment. We recognized that the information would be very helpful to understand carbon allocation within root systems. Instead, as you commented, we added some relevant discussion about more detailed measurements in the discussion section. Please find the line 358-263. Also, we plan to measure those components next year. Although we could not include those detailed measurements into this paper, it will be very worth to be timely published. Hopefully you agree with this. Thank you very much for your valuable comments again.
Line 358-363: “In our current study, short-term drought stress in summer may not increase root to shoot ratio for L. kaempferi seedlings as the drought could alter root structures by allocating more carbon to 1st- and 2nd-order fine roots than coarse roots [73]. Thus, further studies on drought responses to root morphological and physiological traits such as mean root diameter, specific root length and respiration will be prerequisite to understand biomass allocation for various species under a changing climate.”

Round 2
Reviewer 1 Report
Please, in the attached pdf file you can find the review report.